# Post-Traumatic Stress Disorder among Survivors of the September 11, 2001 World Trade Center Attacks: A Review of the Literature

**DOI:** 10.3390/ijerph17124344

**Published:** 2020-06-17

**Authors:** Meghan K. Hamwey, Lisa M. Gargano, Liza G. Friedman, Lydia F. Leon, Lysa J. Petrsoric, Robert M. Brackbill

**Affiliations:** World Trade Center Health Registry, New York City Department of Health and Mental Hygiene, Long Island City, NY 11101, USA; mhamwey@health.nyc.gov (M.K.H.); lisa.gargano@gmail.com (L.M.G.); lfriedman1@health.nyc.gov (L.G.F.); lleon1@health.nyc.gov (L.F.L.); lsilverstein@health.nyc.gov (L.J.P.)

**Keywords:** posttraumatic stress disorder, 9/11, survivors, World Trade Center, cormorbidities, review

## Abstract

Prior reviews of 9/11-related post-traumatic stress disorder (PTSD) have not focused on the civilian survivors most directly exposed to the attacks. Survivors include those individuals who were occupants of buildings in or near the World Trade Center (WTC) towers, those whose primary residence or workplace was in the vicinity, and persons who were on the street passing through the area. This review reports published information on the prevalence of and risk factors for PTSD, as well as comorbidities associated with PTSD among 9/11 survivors. Articles selected for inclusion met the following criteria: (1) full-length, original peer-reviewed empirical articles; (2) published in English from 2002–2019; (3) collected data from persons directly exposed; (4) adult populations; and (5) focused on non-rescue or recovery workers (i.e., survivors). Data were extracted with focus on study design, sample size, time frame of data collection post-9/11, PTSD assessment instrument, and PTSD prevalence, risk factors, and comorbidities. Our review identified the use of cross-sectional and longitudinal designs, finding multiple direct comorbidities with PTSD, as well as the prevalence and persistence of PTSD. Future research would benefit from incorporating more mixed methods designs, and exploring the mediating mechanisms and protective factors of the known associations of PTSD among the 9/11 survivor population.

## 1. Introduction

Among human-made disasters, terrorist events, and mass murders, the attack on the World Trade Center (WTC) on September 11th, 2001 (9/11) stands out as an event resulting in nearly 3000 deaths on the day of the attacks, as well as extensive long-term health effects that continue to be documented 19 years later. In the years since the 9/11 disaster, post-traumatic stress disorder (PTSD) among individuals directly or indirectly traumatized by the attacks has emerged as an important adverse health outcome [1]. In general, PTSD is the most salient category of psychopathology experienced by those traumatized by large scale disasters, including 9/11. This mental health disorder can also result in both substantial individual and population burden of comorbid psychological and physical conditions [2,3]. Given the potential impact 9/11 had and continues to have for those affected, it is important to understand the current research efforts as they pertain to PTSD, as well as the often-underrepresented group of 9/11 “survivors”. Survivors include individuals who were occupants of buildings in the WTC complex, in the vicinity of the attacks (but not first responders), those whose primary residence was in the vicinity of the attack, children, and persons who worked in the area but who were not necessarily present on the morning of 9/11.

### 1.1. Post-Traumatic Stress Disorder

PTSD is defined as the experience of a traumatic or terrifying event that is usually outside of one’s normal daily routine, often characterized by intrusive and distressing memories of the initiating trauma, difficulty in relationships, sleep disruptions, hypervigilance or overreactions to perceived signs of danger, and avoidance or withdrawal from human interactions [4]. PTSD can prevent someone from participating in typical social activities including working and leisure, and can result in negative health behaviors such as the excessive consumption of alcohol or other psychoactive substances [5,6,7]. In the case of the 9/11 attacks, thousands of individuals were directly exposed to the terrorist event. Traumatic experiences for both survivors and rescue and recovery workers included evacuating the towers and surrounding buildings, witnessing people jumping or falling from the towers, being caught in the massive dust/debris cloud from the collapsing towers and nearby buildings, and being a first responder witnessing the death of co-workers. 

### 1.2. Historical Context of PTSD

Although post-traumatic stress (PTS) like symptoms have been reported primarily in relation to war and combat related trauma, disaster-related PTS has been a focus of mental health practitioners since the late 1970’s [8]. Findings from the National Vietnam Veterans Readjustment Study (NVVRS) in the late 1980’s recognized PTS as a disorder and a substantial societal problem. In a controversial report to Congress in 1988, it was estimated that in a sample of 1200 veterans, 15% of those with combat exposure were currently suffering from PTSD and that 30% had developed PTSD during their lifetimes [9]. A follow-up subgroup analysis of the NVVRS data suggested a strong correlation between history of military service and self-reported accounts of stressful combat experience, as well as a concomitant association between level of military service history and a clinical diagnosis of PTSD [10]. It is notable that approximately 10% of veterans suffered from and were impaired by current PTSD more than 10 years after their wartime experiences had ended, suggesting that PTSD symptoms can be longer lasting.

### 1.3. PTSD Prevalence and Risk Factors

Prior to the 9/11 terrorist attacks, much of the previous research which involved civilian populations focused on natural disasters, such as the Buffalo Creek dam collapse in 1972 resulting in the deaths of 125 people with a prevalence rate of 44% with probable PTSD after two years. Another example is the Mt Saint Helens volcano eruption in 1980. Studies conducted within two years after the disaster found that among a sample of community members, 21% of women and 11% of men were afflicted with PTSD or related conditions [8,11]. Importantly, research suggests that man-made disasters such as terrorist attacks are typically associated with higher rates of PTSD compared with natural disasters (human-made disaster PTSD rates: 25% to 75%+ vs. natural disaster PTSD rates: 5% to 60%) [8,12,13,14]. Further, those individuals who are closer to a disaster, or even directly exposed, tend to have a greater likelihood of experiencing PTSD symptoms, whereas those who are first to respond to the disaster are at a lower, albeit still significant, risk of experiencing PTSD symptoms [12]. Those who are farther from the disaster (e.g., the general population), tend to have the lowest likelihood of experiencing symptoms of PTSD; however, these individuals may still report symptoms [12]. Critically, although research has identified PTSD as a substantial factor impacting individuals post-disaster, these findings have often been limited to cross-sectional designs, hindering our understanding of PTSD and its role in individual health and wellbeing over time. 

There were several reports on the prevalence of and risk factors for PTSD within six weeks and up to two years after the 9/11 event for both national and New York City (NYC) residents. In one study, a telephone survey of approximately 1000 Manhattan residents who lived south of 110th Street five to eight weeks after 9/11 found that 7.5% of residents had symptoms consistent with probable PTSD and that persons living south of Canal Street (within <1 mile of the WTC site) had a 20% prevalence of PTSD [15]. Thus, proximity to the disaster site appears to play an important role in prevalence of PTSD symptoms. Two additional follow-up cross-sectional surveys (at four months and six months after the 9/11 event) with the same population of Manhattan residents living below 110th Street detailed a considerable decline in prevalence of PTSD down to 0.6% at six months. Of those participants who were in the WTC complex on 9/11, 19% reported experiencing PTSD symptoms six months later [16], suggesting that exposure severity or proximity to the event may be related to PTSD symptomology. Using these findings as a baseline measure of PTSD in the population exposed to 9/11, two additional follow-up surveys were conducted one year apart in the fall and early winter of 2004–2005 [17]. Findings indicated that the prevalence of PTSD three years post 9/11 was 13.6% and increased to 14.3% by the fourth year, suggesting that PTSD not only remains salient, but, in fact, can worsen over time. 

Prior reviews of 9/11-related PTSD have not focused on persons directly exposed to the attacks (i.e., survivors) and as such potentially at higher risk for this outcome. This may be the case given most civilian survivors have less training or experience compared with responders, in turn predisposing them to potential mental health burdens. Exposures survivors experienced included but were not limited to presence at the disaster area on 9/11, and witnessing the towers being struck and/or collapsing. One review, for instance, provided results of studies published between 2001 and 2008 that focused predominately on the mental health issues and substance use of first responders and volunteers [18]. Another review provided information pertaining to broader populations such as community studies, rescue/recovery workers, Pentagon staff, WTC evacuees, NYC workers, primary care patients, as well as children and adolescents [17]. Importantly, this review did not assess the longer-term patterns of PTSD in these populations. A more recent review of 9/11-related PTSD [19] provided prevalence and correlates of PTSD over time among first responders and 9/11 workers; however, the population for this study was mixed and included some first responders and 9/11 workers so that it is not possible to ascertain from this study the impact of the 9/11 disaster specifically for survivors.

### 1.4. Study Aims

This review sought to identify studies focused on the “survivor” population from the attacks on 9/11. Survivors include those individuals who were occupants of buildings in or near the WTC complex, in the vicinity of the attacks but were not first responders, those whose primary residence was in the vicinity of the attack, and people who may have been passersby on the day of the attacks. Using these criteria, this review focused on the prevalence of and risk factors for the initiation and persistence of PTSD, emerging co-morbidities, and ameliorating factors specific to survivor groups. 

## 2. Methods

### 2.1. Data Sources and Searches

Relevant studies were identified by searching PubMed and PsycINFO for all published articles up to January 2020, using the relevant search term (“WTC” OR “World Trade Center#x201D; OR “September 11, 2001#x201D; OR “September 11#x201D; OR “9/11#x201D;) AND (“PTSD#x201D; OR “post-traumatic stress disorder#x201D; OR “posttraumatic stress disorder#x201D;). Published review articles on 9/11-related PTSD were also examined to identify additional relevant publications.

### 2.2. Inclusion/Exclusion Criteria

Selection criteria included: (1) full-length, original peer-reviewed empirical articles; (2) published in English from 2002–2019; (3) collected data from persons directly exposed to the 9/11 disaster in NYC (e.g., physical proximity to the New York City disaster area on 9/11 or the aftermath); (4) focused on adult populations; and (5) focused on non-rescue/recovery/clean-up workers (Rescue and Recovery Workers = RRW) or contained analyses separating RRW from non-RRW. Although classified as a survivor population, manuscripts pertaining specifically to children were not included in this analysis because PTSD was not consistently measured or evaluated within this population. 

### 2.3. Study Selection

All relevant titles, abstracts, and papers (*N* = 491) were reviewed independently by at least two researchers. After the initial screening by two researchers (LF and LL), 80 abstracts were identified as potentially relevant and reviewed by four researchers (LG, LP, LF, and LL), and 47 were selected for inclusion. A subsequent, more detailed screening of the 47 abstracts was conducted, resulting in the exclusion of 17 articles for failing to meet inclusion criteria. Thus, a total of 30 articles were identified as meeting inclusion criteria and subsequently included in this review (Figure 1). 

### 2.4. Data Extraction

Data relevant to 9/11-related PTSD were extracted and examined by all researchers. Data extracted included study design and sample size, time frame of data collection post-9/11, PTSD assessment instrument/criteria, and, when applicable, PTSD prevalence, risk factors, and comorbidities (Table 1).

## 3. Results

### 3.1. Data Source: The World Trade Center Health Registry 

Many of the articles reviewed used data from the World Trade Center Health Registry (Registry). Briefly, Registry data comprise a longitudinal cohort of rescue/recovery workers and volunteers (RRW), as well as community members who were not involved in the rescue/recovery efforts (i.e., residents and children enrolled in schools in lower Manhattan, occupants of destroyed or damaged buildings, and passersby in the vicinity of WTC). At Wave 1 (September 2003–November 2004), 71,431 participants completed a computer-assisted or in-person baseline enrollment interview. Three additional follow-up waves of data collection have been completed since baseline, resulting in four waves of available data (Wave 1 = 2003; Wave 2 = 2007; Wave 3 = 2011; Wave 4 = 2015). These surveys focused on health information; specifically, the enrollee’s medical history and both physical and mental health status, including measures specific to PTSD [1]. Other datasets were used in some of the reviewed articles and are detailed in the subsequent results and Table 1. 

### 3.2. Early Reports of PTSD Prevalence among 9/11 Survivors

There were several attempts made by researchers shortly after 9/11 to establish PTSD prevalence among Manhattan residents living south of 110th Street and various subgroups. PTSD prevalence in these studies ranged from 7.5% to 11.2% [20]. The Registry baseline survey (2003–2004) reported a PTSD prevalence of 13.2% among lower Manhattan residents, 16.9% among lower Manhattan office workers, and 19.3% among passersby, which increased to 16.3%, 19.1%, and 23.2%, respectively (i.e., approximately 3% increase per group), at the first follow up survey (2006–2007) [1]. Early estimates of PTSD among Manhattan residents below 110th Street were obtained at one to nine months post-9/11. Among people who lived south of 14th Street in Manhattan, for instance, PTSD prevalence was 12.3%, compared to 7.2% for those who lived north of 14th Street and south of 110th Street [16], suggesting proximity effects. Also, a study conducted five to eight weeks after 9/11 with 988 adult Manhattan residents living south of 110th Street on 9/11 reported a slightly higher PTSD prevalence of 9.9% in women, compared to 4.8% in men [21]. Among participants without a pre-9/11 diagnosis of PTSD, the prevalence of PTSD at Wave 4 (2015–2016) was 15.7% for passerby, 11.9% among area residents, and 13.7% for area workers [22]. Although not explicitly tested, these results suggest a small decline compared with baseline percentages [16,20]. 

### 3.3. Cross-Sectional Findings Pertaining to Survivors

Cross-sectional studies have often been utilized as a means of understanding PTSD prevalence among the 9/11 survivor population. Of the 25 articles reviewed, 11 used a cross-sectional design to evaluate PTSD among 9/11 survivors. 

Two studies of Chinese immigrants [23,24] presented findings from surveys conducted at two time periods (May 2002 and March 2003) with 65 garment workers who lost their jobs as a result of 9/11. One study assessed mental health status—both PTSD and depression—and found a PTSD prevalence of 21% at eight months after 9/11; 27% met the diagnostic criteria for PTSD at 18 months. Further, those who received a PTSD diagnosis at either timepoint were found to have higher severity of symptoms at both times than those with Time 1 or Time 2 only PTSD. These results suggest that over time the emotional impact of a traumatic event may become more salient for some, particularly those who report symptoms of PTSD soon after event exposure. 

A study of mothers who were directly exposed to 9/11 and recruited through childcare centers evaluated PTSD symptom clusters and comorbid depression four years post-9/11 [25]. Mean scores were calculated for each of the three symptom clusters, avoidance (M = 2.0, SD = 2.88, range = 0–13), arousal (M = 2.38, SD = 3.05, range = 0–14), and re-experiencing (M = 2.56, SD = 2.99, range = 0–13), suggesting moderate levels of PTSD symptomology severity within this group four years later.

Eight papers provided PTSD prevalence from studies using Registry enrollees. An analysis of the entire Registry survivor population found an overall prevalence of 16.3% two to three years after 9/11; prevalence was highest (19.4%) among building occupants and passersby in lower Manhattan on 9/11 [26]. Another study using Registry data [27] focused only on area residents and used three methods to assess prevalence, including a DSM-IV diagnostic criteria, a cutoff score of 44 on the PCL, and a combination of both diagnostic criteria and cutoff score. The DSM-IV diagnosis criteria and a PCL cutoff yielded similar prevalence of 16.1% and 15.1%, respectively. In a separate analysis of Registry data which focused on WTC tower survivors two to three years post-disaster, there was an overall PTSD prevalence of 15% [28]. A follow-up paper on tower survivors found late onset PTSD (i.e., first reported at Wave 3) prevalence for tower survivors in 2010 and 2011 to be 4.5% and 4.3% for survivors overall, respectively [29]. However, the prevalence of chronic PTSD was 13.6% among those in Tower 1 and Tower 2, compared to 10.3% for enrollees in other buildings, and 9.3% for passersby [29]. One study focused on the Asian community using Registry data collected two to three years post-9/11 found a PTSD prevalence of 14.6% [30], while another study of Asian Americans 11 to 12 years post-9/11 found a prevalence of 15.2% among non-Rescue/Recovery workers [31]. 

### 3.4. Longitudinal Findings Pertaining to Survivors

This review found 11 papers presenting results of longitudinal studies on PTSD in survivor populations. Within survivors, the groups represented by these longitudinal studies included persons who were designated as being highly exposed, in the WTC towers on 9/11, or members of a specific population such as residents of lower Manhattan, area workers, or persons of Asian descent. In addition, other longitudinal studies evaluated co-morbid depression, respiratory problems, and reported levels of PTSD in relation to these conditions. For instance, one study [32] assessed a highly-exposed sample of survivors who were in the WTC towers or who were within several blocks of the WTC on 9/11 (*n* = 45), reporting that PTSD symptoms were modest (M = 16.71, SD = 10.78) at seven months post-9/11. Mean PTSD levels decreased only marginally by approximately a quarter of a point at follow up 15 months later. In a separate report on this same sample stratified by gender, there was a PTSD prevalence of 16% for men and 13% for women at seven months post-9/11, though women had a higher prevalence of PTSD at follow up at 18 months [33], suggesting potential ameliorating factors for men that did not occur with women. 

A number of reports focused on persons who, by definition, were highly exposed by being occupants of the WTC towers before and during the attacks. One Registry based report of over 1300 survivors in or nearby the towers indicated that 13% of participants had probable PTSD in 2015-2016; a slight decrease from 16.5% in 2003–2004 [34]. A separate report on tower survivors [35] using Registry data reported a consistently high PTSD prevalence of 18.0% (2003–2004), 20.5% (2006–2007), and 18.1% (2015–2016). This study further stratified results by gender finding that women had a higher prevalence of PTSD than men at each time point [35]. 

In regard to lower Manhattan area workers and residents, a longitudinal study of patients enrolled in the WTC Environmental Health Center (EHC) at Bellevue Hospital [36] examined baseline enrollment data (August 2005–February 2009) compared with data from a subsequent monitoring visit (between October 2009 and May 2016). Given that this study focused on a patient population, the prevalence of PTSD was even higher than building survivor studies cited above, with a baseline PTSD prevalence of 53% among women and 46% among men. At the follow-up visit, 51% of women who identified with PTSD at baseline had persistent PTSD, while 49% of men who reported PTSD at baseline had persistent PTSD at follow up [36]. 

A longitudinal study with the lower Manhattan Asian community compared outcomes with Whites living in the downtown area (Asian New Yorkers n = 2431; vs. 31,455 White New Yorkers). Reviewing Registry data two to three and five to six years post-9/11, Asian New Yorkers were found to have a higher proportion of probable PTSD in the delayed onset (i.e., developing years after 9/11) group (8.6% vs. 7.4%) and remitted or recovered (5.9% vs. 3.4%) group [30]. 

A case-control study of survivors at two time points (2008–2010 and 2013–2014) examined the course of Lower Respiratory Symptoms (LRS) and corresponding PTSD using Registry data. Among enrollees without LRS, 2.5% had PTSD, those with resolved LRS had 29.1% PTSD, and those with persistent LRS had a 43.6% PTSD prevalence [37], suggesting that PTSD and LRS are associated, though directionality is still being evaluated. 

### 3.5. Risk Factors for PTSD

Studies demonstrated that several factors substantially increased the risk of PTSD after 9/11, including gender, age, race/ethnicity, education, and income. For instance, several studies indicated that a greater proportion of females reported symptoms of PTSD compared with their male counterparts [27,28,38] while another study of displaced Chinese workers in NYC found no gender associations to PTSD risk [23]. Age associations were mixed, with one study reporting that being older in age was often concurrent with greater risk of PTSD among tower survivors [28], while another [35] found that being younger in age on 9/11 was associated with higher PCL scores. However, Race/ethnicity has been consistently associated with PTSD, such that higher PTSD prevalence was associated with non-white racial groups [27,28], and Latino ethnicities [34]. Finally, several studies have identified marital status [27] and lower educational attainment [27,35,39], as being associated with increased risk for PTSD. Additional socioeconomic factors, such as lower income [27,28] and unemployment [35,39] were also associated with PTSD. In particular, financial concerns among displaced Chinese workers in NYC were positively correlated with the number of PTSD symptoms. [23].

In a study of 36,897 survivors [22], PTSD prevalence was higher among participants who experienced specific types of 9/11-related exposures compared with those who had not. Examples of 9/11-related exposures included presence in the dust cloud, sustaining a 9/11-related injury, and personally witnessing traumatic events like an airplane striking a WTC tower or someone being injured or killed. Exposure characteristics were further associated with subsequent PTSD, including fear for safety [38,39] and witnessing horrific events on 9/11 [1,27,28,29,35,39,40], such as the death or injury of others [38]. Overall higher 9/11-exposure [28,34], sustaining an injury on 9/11 [1,27,28,29,35,39], being caught in the dust cloud from collapsing buildings [1,27,28,29], bereavement [1,39], and post-9/11-related job loss [1,29,39] were all associated with subsequent PTSD. 

In a longitudinal study of 1304 WTC tower survivors [34], participants with PTSD were found to have significantly greater exposure to the events of 9/11 than those without PTSD. Studies among persons who evacuated the WTC towers and/or other damaged or destroyed buildings have found a relationship between PTSD and longer evacuation time [28], such that the longer it took to evacuate, the greater the risk of PTSD. Further, having worked for an employer that sustained fatalities [28], later evacuation initiation [28,34], encountering infrastructure and/or behavioral barriers during evacuation [29,35], and being above the impact zone of the airplane [28], were all associated with greater risk for PTSD. 

Studies on lower Manhattan residents found a dose-response relationship between evacuation/time displaced from residence and odds of PTSD [27]. Additionally, evacuating one’s home, longer time required to be away from home [26], and damage to one’s home, with or without a heavy layer of dust [1], were associated with PTSD. A further analysis of survivors, specifically office workers in lower Manhattan found that those who evacuated, had damage to their workplace with or without a heavy layer of dust, and earlier time returning to work were more likely to have PTSD [1]. Other risk factors have been shown to be associated with PTSD risk among survivors, including lower social support [1,34,35,39], unmet mental healthcare needs [34,35,39], pre-9/11 mental health diagnosis [1], loss of possessions [38], lower-self-efficacy [34], and poor quality of life [39].

Persistence of PTSD symptoms have also been found to be problematic for 9/11 survivors. In one study [39] evaluating the trajectory of PTSD experiences for survivors over time with Registry data, individuals with the most direct exposure (i.e., in the vicinity of the disaster) reported persistent or worsening symptoms from baseline (2003–2004) to Wave 3 (2011–2012). Additional factors were also identified as being associated with persistent PTSD symptoms, including sustaining an injury on 9/11, fearing for one’s safety, low social integration (e.g., few close friendships, no community engagement), job loss related to 9/11, and unemployment [39]. 

### 3.6. Comorbidities

Depression is the most commonly reported co-morbid condition present with PTSD. In a study examining the longitudinal determinants of depression 14 to 15 years after 9/11 [41], of the 27% of lower Manhattan area workers who ever screened positive for probable PTSD (≥44 on PCL-17) fgv b since 2003, 53% met the criteria for PHQ-8 depression. Similar levels of comorbid PTSD and depression were reported for lower Manhattan residents, with 50% meeting the criteria for depression among the 21% who ever screened positive for probable PTSD. Of those lower Manhattan area workers or residents who reported no PTSD, only 5% had depression. 

A study [25] which examined the association between PTSD symptom clusters and depressive symptom severity among a sample of mothers directly exposed to the WTC attacks reported a significant correlation between high arousal cluster scores and chronic depression. The association of PTSD and chronic depression was also examined in a longitudinal study [32] exploring the effect that individual differences in adult attachment style had on highly exposed adults’ adaptation to the events of 9/11 at seven months and 18 months post-disaster. Securely attached adults exhibited the lowest levels of PTSD, which decreased over time, while there were no main effects of attachment-related anxiety or avoidance on depressive symptom levels or change in symptom levels over time. 

Associations between PTSD and depressive symptoms have been further shown to be related to prescription drug use. Specifically, two cross-sectional studies of Chinese immigrant workers [23,42] found PTSD symptom level and depression were both associated with beginning or increasing use of prescription drugs as a result of 9/11. Those who either began using prescription medications after 9/11 or increased their medication usage after 9/11 exhibited higher mean PTSD symptom levels than those who did not. 

In addition to depression, gastroesophageal reflux syndrome (GERS) was frequently highlighted as a comorbid physical condition with post-9/11 PTSD among survivors. One study [43] examined the incidence of persistent GERS among area workers, residents, and passersby in relation to 9/11 dust cloud exposure levels (categorized as “none”, “some”, or “intense” dust cloud exposure) within two comorbid strata: those with PTSD alone and those with PTSD and asthma. A strong dose-response gradient was found for the incidence of persistent GERS with comorbid PTSD in relation to 9/11 dust cloud exposure level. The cumulative incidence of persistent GERS among participants with co-morbid PTSD and asthma was 23% among those with no dust cloud exposure, 32% among those with some dust cloud exposure, and 34% among those with intense dust cloud exposure. The cumulative incidence of persistent GERS among participants with PTSD alone (no comorbid asthma) was slightly lower, with 17% for those with no dust cloud exposure, 22% among those with some dust cloud exposure, and 26% for those with intense dust cloud exposure. A follow-up study [44] examined new onset GERS post-9/11 and persistent GERS and its association with comorbid asthma and PTSD. This study found a 50% increased risk for persistent GERS associated with early post-9/11 PTSD (2003–2004) among survivors (defined in this report as community members exposed to 9/11 attacks), suggesting a strong comorbid association between GERS and early PTSD among 9/11 survivors.

Lower respiratory symptoms (LRS) have also been shown to be comorbid with PTSD for 9/11 survivors. In particular, three studies examined the association of LRS (e.g., shortness of breath, wheezing or cough) and comorbid PTSD. In a case-control study on the risk factors for persistence of LRS among survivors of the 9/11 attacks [37], results indicated that PTSD was associated with persistent LRS. Specifically, within the study population 2.5% had PTSD and no LRS, 29% had PTSD and resolved LRS; 44% had PTSD and persistent LRS at the second time point of data collection, in 2013–2014. The adjusted odds ratio of PTSD associated with persistent LRS at the follow up exam compared with resolved LRS was 2.8. These results suggest a bidirectional exacerbation of symptoms between LRS and PTSD. In a longitudinal study of co-occurring LRS and PTSD five to six years after 9/11 [45], findings indicated a close association between the two conditions, independent of shared risk factors. Participants with LRS were 4.2 times more likely to report PTSD, while participants with PTSD were 4.3 times more likely to report LRS. Those with comorbid PTSD and LRS had poorer quality of life than those with either condition alone or those with neither condition. Participants with comorbid PTSD and LRS were more likely to report a new medical diagnosis of depression or anxiety, have symptoms of severe nonspecific psychological distress, and report medication use for a mental or emotional condition than those with PTSD alone. As more comorbid associations are identified, it is likely additional factors may compound these effects for 9/11 survivors. A third longitudinal study used Registry data to conduct a mediation analysis evaluating the bidirectional associations of 9/11 exposure on PTSD and LRS. Findings suggested that among community members, dust cloud and high psychological exposures were more strongly associated with probable PTSD six to 10 years post-9/11 [46]. Further, lower respiratory symptoms (LRS) were shown to contribute to the persistence of PTSD, above and beyond the effects of 9/11 exposure and individual characteristics [46]. In combination, these results indicate that although probable PTSD should be considered a risk factor for the development of LRS, LRS may also be a predictor of PTSD, thus suggesting a bidirectional influence process. 

Finally, chronic physical health conditions have been found to be comorbid with PTSD among 9/11 survivors and their decisions to retire early. One study [47] examined the impact of chronic physical health conditions, with and without comorbid PTSD, on early retirement and income loss among the survivor population of the WTC Health Registry. Though having PTSD alone was not found to increase the odds of an individual’s early retirement (AOR 1.2, 95% CI 0.9–1.6), among those with three or more physical health conditions and PTSD, the odds of early retirement increased in a dose-response manner (AOR 3.4, 95% CI 2.4–4.7).

## 4. Discussion

This review consolidated information on PTSD for a defined population of survivors of the 9/11 disaster. An initial search of the literature yielded nearly 500 studies but after culling these articles only 30 presented information specifically for survivors. Despite the paucity of research on survivors and PTSD post 9/11, the available evidence indicates that this group was critically affected by the tragedy. Further, PTSD is often co-occurring with other physical and psychological factors increasing the health burden and reducing quality of life considerably for these individuals. 

The focus on the 9/11 survivor cohort presents researchers with a unique perspective that will help inform future studies; however, despite the efforts of the presented studies, many gaps still exist. For instance, the identification of potential protective factors associated with the development of PTSD over time has received limited attention in these studies. What remains unknown is why PTSD symptoms increase for some over time and decrease for others, and what additional mechanisms may be impacting these variations. 

Further, it is necessary to consider the degree to which prevalence of PTSD among 9/11 survivors is influenced by the multiplicative effects of exposure. Across the studies reviewed, a multitude of 9/11-related exposures were used to categorize the experiences of survivors. Often these exposures included presence in the dust cloud, sustaining a 9/11-related injury, and personally witnessing traumatic events like an airplane striking a WTC tower or someone being injured or killed. Other types of exposures may also be referenced in these studies, such as specific locations, evacuating nearby buildings, or other work-related exposures. Importantly, early studies of 9/11-related exposure suggest that the severity and number of exposures experienced directly influence the prevalence rates of PTSD [20]. PTSD prevalence and course over time are determined not only by exposure to the events of 9/11, but also the degree of exposure, as well as the number of exposures. Although both rescue/recovery workers and survivors may have experienced some similar exposures on 9/11, survivors may, in fact, have experienced additional factors, such as evacuating a building, running from the dust cloud, witnessing the planes hit the towers, and/or being injured. These additional exposures make survivors a critical group to further consider in terms of how their unique experiences impact their PTSD prevalence over time. 

The persistence of PTSD and its predictors are critical for researchers to consider and further evaluate. Of the limited longitudinal studies evaluating PTSD persistence, some research suggests that persistent PTSD is predicted by 9/11 exposure severity and 9/11-related job loss [39]. Further, this review identified a consistent gender association among survivors. Specifically, women were more likely to report persistent PTSD over time compared with their male counterparts. Among survivor groups, research suggested women reported higher rates of PTSD over time, whereas among the rescue and recovery groups, men tended to report greater PTSD [48]. Importantly, these differences may be accounted for by the fact that survivor groups have a larger proportion of women compared with the rescue and recovery and recovery groups. Despite these longitudinal findings, it is not well understood what factors may be ameliorating the effects of PTSD for males compared with women, nor is it clear what potential additional factors may be driving this persistence. Importantly, in the general, non-9/11 exposed population, women are two to three times more likely to report experiencing symptoms of PTSD during their lifetime. When considering these gender discrepancies and how potential long-lasting implications, one potential factor may be that women tend to report greater utilization of mental healthcare services compared with men. This, in turn, increases their likelihood of maintaining a PTSD diagnosis, while simultaneously receiving treatment to attempt to resolve the issue. 

The persistence of PTSD in survivors may also be related to presence of complicated grief (CG) due to the death of someone close. CG has only recently been recognized as a mental health disorder and is probably underdiagnosed and undertreated within the 9/11 bereaved survivors. Persons with this condition are likely to have a significant degree of functional impairment and intransigence to traditional treatments. Further research on CG among survivors especially those with PTSD would be important for establishing the basis for developing interventions to ameliorate unresolved effects of CG [49].

Although studies have addressed gender and race/ethnicity as being a potential risk factor for experiencing PTSD, it may be more relevant to further consider the implications of structural inequities as they exist in a stratified society. Those experiencing PTSD symptoms may been even more likely to experience these symptoms when structural inequities and barriers exist that ultimately compound stress and, in turn, may worsen the effects of PTSD. Future studies should consider these inequities and how 9/11-related PTSD may be subsequently impacted by them.

Barriers also exist to assessing and understanding PTSD persistence among 9/11 survivors. Studies have used different definitions and exposure measures, making comparisons between studies challenging. Through using a more standardized definition of exposure and its classification categories for survivors of 9/11, researchers will obtain a more complete understanding for why PTSD manifests and then persists after a catastrophic event. 

It must also be noted that the measures, techniques, and thresholds used to assess PTSD are often varied and not consistent across studies. Even with the most frequently used measures (i.e., PCL and DSM-IV diagnostic criteria (i.e., cutoff score of 44) there are inconsistencies across studies. Although future research would benefit from identifying and implementing a consistent method of PTSD assessment, we recognize the challenges to doing so (e.g., clinical implications, measurement differences, longitudinal data collection). To help alleviate these measurement constraints, additional methodological and design steps could be taken to improve our understanding of PTSD among survivors and all other groups. Specifically, many studies use a purely quantitative design to assess PTSD in post 9/11 survivor populations. Although this method provides essential data for understanding the links between a variety of factors and post 9/11 PTSD, it may be missing critical details. As such, it would be beneficial for future research to consider a mixed method approach whereby qualitative data are incorporated into the research design and results. In doing so, researchers may be able to gain further insight into the additional factors that may be exacerbating or ameliorating PTSD for survivors over time.

Further, studies of military veterans point to the importance of understanding and providing care for those individuals with subthreshold PTSD, or those who report clinically significant PTSD symptoms, but do not meet the full diagnostic criteria for a PTSD diagnosis [50]. Importantly, among those with subthreshold PTSD, elevated depression and suicidal ideation have been identified [51,52], as well as greater alcohol use [53], anger and aggression [54], fewer days worked [55] and lower social and family functioning [56]. Overall, those with subthreshold PTSD report increased overall functional impairment [57]. Given the comorbid associations between subthreshold PTSD among veterans, it would be useful for future researchers to further consider these physical and mental health characteristics as they apply to 9/11-exposed survivors. 

Comorbid conditions are also not well understood among the 9/11 survivor population. Specifically, much of the research focuses on mental or physical health and does not attempt to understand the two in conjunction. In fact, many studies suggest that individuals who experience mental or physical health symptoms who simultaneously report PTSD symptoms are experiencing comorbid conditions [23,32,37,42,43,46]. Given the potential bi-directional implications of PTSD symptoms and risk, it is necessary to consider both the mental and physical implications that may co-occur with PTSD symptoms over time. In fact, much of the research seems to suggest that a comorbidity would potentially exacerbate symptoms of PTSD, or other conditions. Future research would benefit from more adequately addressing these comorbid associations and determining their causal links. 

To better understand the impacts of PTSD for the 9/11 survivor population researchers must focus on the potential mediating mechanisms that can perpetuate or improve these symptoms. For instance, it may be that those individuals who were directly exposed to 9/11 but had access to mental health services, community support programs, and other such aid, experienced diminished PTSD. Additionally, the assessment of prior exposure to trauma is often missing from evaluations of current PTSD symptoms and traumatic exposures. In fact, it may be that the individuals who report currently experiencing symptoms of PTSD may have additional traumatic exposures beyond their 9/11 experience. For instance, a childhood trauma may increase vulnerability to 9/11-related PTSD, or a trauma experienced post-9/11 (e.g., Hurricane Sandy) may have exacerbated the experiences of 9/11 and thereby increased PTSD symptoms [58]. As is typically the case in this type of research, it is difficult to determine the temporal sequencing of these experiences, as well as their impact on current PTSD symptoms. 

## 5. Limitations

Although we were able to identify and review 30 empirical articles addressing PTSD among 9/11 survivors, additional articles may exist that were not able to be included in this sample. Specifically, some studies using a 9/11-exposed population do not distinguish between rescue and recovery workers and survivors, making it challenging to disaggregate findings between the two groups. Further, because of the limitations in measuring PTSD among children, we were unable to include them in this study, limiting our ability to understand and evaluate the impact of 9/11 on children after 9/11. 

## 6. Conclusions

Through conducting this systematic review, we have identified several areas of research that are currently being conducted in relation to PTSD and the survivor population of 9/11. Specifically, research has incorporated cross-sectional and longitudinal designs; however, a mixed-methods approach is still needed. Further, multiple direct and comorbid associations with PTSD have been identified, including exposure/vicinity, depressive symptoms, GERS, LRS, prescription drug use. Additional demographic risk factors including age, race/ethnicity, and gender may act as proxies for structural inequalities. Additionally, the prevalence and persistence of PTSD among 9/11 survivors continues to be a critical area of research. What remains largely unstudied are the mediating mechanisms that link these associations, as well as the protective factors that may help alleviate PTSD for 9/11 survivors. 

## Figures and Tables

**Figure 1 ijerph-17-04344-f001:**
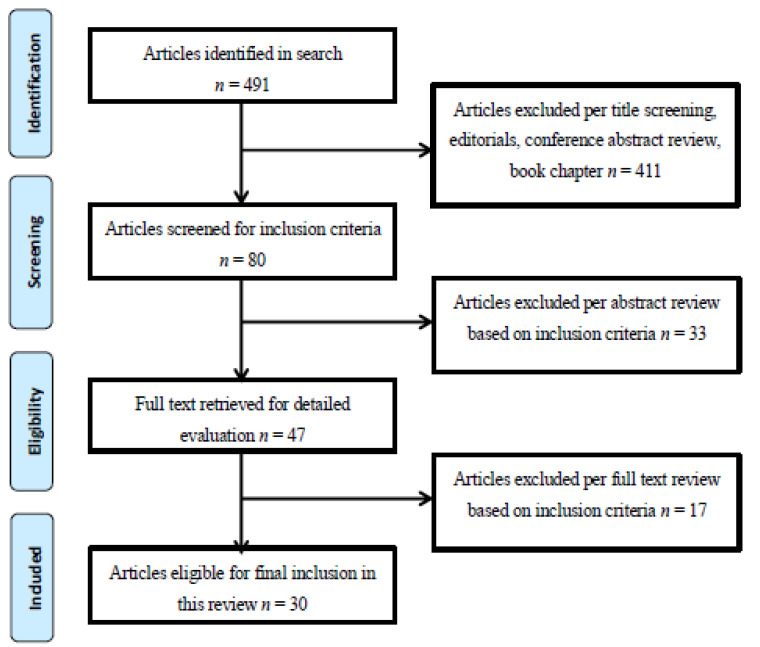
Method for identification screening and inclusion/exclusion in the review.

**Table 1 ijerph-17-04344-t001:** Data extracted from all included studies. PTSD: post-traumatic stress disorder; SD: Standard Deviation; LRS: Lower Respiratory Symptoms; GERS: Gastroesophageal Reflux Syndrome; PCL: PTSD Checklist; AOR: Adjusted Odds Ratios.

Author, Year	Study Design	Sample Size	Time Frame of Data Collection Post-9/11	PTSD Instrument	PTSD Prevalence	PTSD Correlates (PTSD is Outcome)	PTSD Comorbidities (PTSD is Covariate)
Adams et al., 2018	Longitudinal	1304	Four time points: 2003-04; 2006-07; 2011-12; 2015-16	PCL-17 - ≥44 cut-off score	13% PTSD (4.1% PTSD alone)	Greater exposure to the events of 9/11; lower self-efficacy; late evacuation from building; race/ethnicity; post-9/11 life-threating events	
Bonanno et al., 2005	Longitudinal	52	7 and 18 months	PTSD Symptom Scale, Self-Report version (PSS-SR) (cut off score = 28)	7-month mean score 16.42 (SD 10.44) 18 month mean score 13.76 (SD 10.50)	7-month only: witness death or injury, self-enhancement 18-month only: 7-month PTSD score Both time points: Gender, physical danger on 9/11, social constraints	
Bowler et al., 2017	Longitudinal	1755	Three time points: 2003-04; 2006-07; 2011-12	PCL-17 Civilian	Overall 2003-04:18.0% 2006-07: 20.5% 2011-12: 18.1%	Being younger on 9/11, lower education, being unemployed, having experienced more injuries, more evacuation problems, witnessed more horrific events on 9/11, having unmet mental health care needs, and less social support all predicted higher PCL scores at 2011-12 time point	
					Men: 12.1; 17.3; 13.7		
					Women: 26.1; 24.9; 24.1		
Brackbill et al., 2009	Longitudinal	46,322	2003–2004 2006–2007	PCL-17 Civilian	23.8% (95% CI, 23.4%–24.2%) at either time point	Pre-event mental health diagnosis; witnessing horrific events; injury on 9/11; loss/death of other on 9/11; post-9/11 job loss; low social support For residents/office workers: building damage (with and without dust layer)	
					2003–2004: 14.3% [95% CI, 13.9%–14.6%]		
					2006–2007: 19.1% [95% CI, 18.7%–19.5%]		
Brackbill et al., 2014	Longitudinal	14,087	2003–2004 2006–2007	PCL-17 Civilian	*N* = 1586 (11.3%)		The proportion of probable PTSD increased with severity of injury. Probable PTSD was associated with having 9/11-related chronic conditions diagnosed. There was a significant multiplicative interaction between number of types of injuries and probable PTSD and the odds of a chronic health condition.
Caramanica et al., 2014	Longitudinal	29,486	Three time points: 2003-04; 2006-07; 2011-12;	PCL-17 Civilian	15.2% (10.1% PTSD+Depression+ and 5.1% PTSD+Depression-)		
Dekel et al., 2013	Biomarker study	61	T1: Mar 2002 (7 mo >9/11) T2: Apr 2003 (16 mo >9/11)	PSS-SR (PTSD Symptoms Scale Self-Report)	T1: 16.0% (*n* = 13) T2: 13.5% (*n* = 7) met PTSD symptoms criteria, (i.e., reported often or always of at least one reexperiencing, three avoidance, and two hyperarousal symptoms)		In men, PTSD symptoms predicted cortisol response following recollections of their 9/11 experiences.
					Women higher means than men		
DiGrande et al., 2008	Cross-sectional	11,037	2003–2004	PCL-17 Civilian	12.6%	Older age, female gender, Hispanic ethnicity, low education and income, and divorce. Injury, witnessing horror, and dust cloud exposure on 9/11. Post-disaster risk factors included evacuation and rescue and recovery work.	
DiGrande et al., 2011	Cross-sectional	3271	2003–2004	PCL-17 Civilian	15% (*n* = 492)	Gender, race/ethnicity, income. Having been in the towers above the point of airplane impact, having initiated evacuation late, having been exposed to the dust cloud, having witnessed horror, having sustained an injury, and having worked for an employer that experienced fatalities on 9/11. There was a positive relation between the exposure severity and the PCL-S score.	
Farfel et al., 2008	Cross-sectional	71,437	2003–2004	PCL-17 Civilian	16.3%	Residents: evacuated home; early return home	
						Building occupants, passerby, people in transit: being in a damaged or destroyed building, early return to work	
Fraley et al., 2006	Longitudinal	45	7 months (Wave 1) and 18 months (Wave 2)	PTSD Symptom Scale, Self-Report version (PSS-SR) (max score = 51)	Mean (Standard Deviation) Wave 1: 16.71 (10.78) Wave 2: 13.74 (10.87)		Secure adults exhibited healthy adjustment in the months following 9/11
Galea et al., 2003	Longitudinal	1008, 2001, and 2752, respectively	1 month, 4 months, and 6 months	National Women’s Study PTSD module	At 6-month survey— Since 9/11: 6.6% (*n* = 124)		
					Last 30 days among those who developed PTSD since 9/11: 18.4% (*n* = 24)		
Gargano et al., 2016	Longitudinal	7695	Three time points: 2003-04; 2006-07; 2011-12;	PCL-17 Civilian	First time PTSD @W3 4.5% for tower/ building survivors, 4.1% for street.	Being in Tower 1 or 2 (AOR 1.3, 2.0-1.7) Sustaining an injury on 9/11; witnessing 3+ horrific events; caught in the dust cloud; 9/11-related job loss for those in Tower 1 or 2 or other buildings compared to those on the street. Infrastructure and behavioral barriers for building occupants	
					Chronic PTSD 13.6% for Tower, 10.3% other building, 9.3% street		
Gonzalez et al, 2019	Retrospective Cross-sectional	1648	Data from 3 surveys were combined— Leaders In Gathering Hope/Project Restoration (June 2014–August 2016) Sandy/WTC (September 2013–December 2014)	PTSD Checklist PCL-17	Comparison of PTSD rates between responses and community members. Rates of PTSD were much higher among survivors 8.6% vs. 31.1%.		
Huang et al., 2019	Cross-sectional	4721 Asian 42,862 Whites (comparison)	2003–2004	PCL-17 Civilian	Asian -15.1% White- 14.4%	Average number of exposures and PTSD grouping	
						Age Gender Income Marital status Job loss Prior PTSD exposure Status of LRS Social Support Mental Health Treatment	
Jacobson et al., 2018	Longitudinal	21,258	Four time points: 2003-04; 2006-07; 2011-12; 2015-16	PCL-17 Civilian	25.7% ever PTSD at any time point		Depression~ 56% in EVER PTSD group
Jordan et al., 2017	Case-control	545	Exam 1: 2008–2010 Exam 2: 2013–2014	PCL-17 Civilian	Asymptomatic Group 2.5%		Persistent LRS
					Resolved LRS 29.1%		
					Persistent LRS 43.6%		
Jordan et al., 2019	Longitudinal	35,897 WTCHR enrollees	W1 (2003–2004) and W4 (2015–2016)	PTSD Checklist PCL-17	Among participants without a pre-9/11 diagnosis of PTSD, the prevalence of PTSD at Wave 4 (2015–2016) was 15.7% for passerby and 11.9% for area residents and 13.7% for area workers.	PTSD prevalence was higher among study participants who experienced each type of 9/11-related exposure examined than those who had not experienced the respective exposure	More participants reported comorbid PTSD and depression (4.2%) than depression alone (3.0%) and PTSD alone (2.0%).
Li et al., 2011	Longitudinal	37,118	2003–2004 2006–2007	PCL-17 Civilian			Post-9/11 GERS and persistent GERS
Li et al., 2016	Longitudinal	29,406	Three time points: 2003-04; 2006-07; 2011-12	PCL-17 Civilian			Persistent GERS more prevalent among those with PTSD at baseline
							Late-onset GERS also associated with PTSD at baseline
							PTSD mediated the associated between early post-9/11 asthma and late-onset GERS
Nair et al., 2012	Longitudinal	16,363	2003–2004 2006–2007	PCL-17 Civilian Probably PTSD defined as cutoff score of 44 and the presence of at least 1 reexperiencing symptom, 3 avoidance or numbing symptoms, and 2 hyperarousal symptoms	*n* = 2337 (14.3%)		LRS and PTSD are closely associated, independent of shared risk factors. Participants with LRS were 4.2 times more likely to report PTSD. Participants with PTSD were 4.3 times more likely to report LRS.
							Comorbid had lower QoL than those with either condition alone or those with neither condition.
Pulcino et al., 2003	Cross-sectional	988	5–8 weeks	National Women’s Study PTSD module	Women: 8.8% (*n* = 41)		
					Men: 3.6% (*n* = 16)		
Rubacka et al., 2008	Cross-sectional	61 mothers	~4 yrs (30–57 months)	Posttraumatic Stress Diagnostic Scale (PDS)	Mean severity levels of PTSD symptomatology: 2.56 for the re-experiencing (SD = 2.99, range = 0–14), 2.00 for avoidance cluster (SD = 2.88, range = 0–13) and 2.38 for arousal cluster (SD = 3.05, range = 0–14)		PTSD/depression PTSD clusters severity → correlates w/ depression severity
Thiel de Bocanegra et al., 2004	Cross-sectional	77	8 months	PCL-C	Mean 19.3 (SD 12.6) (max 68) PTSD diagnosis (all three clusters met) 21%	Direct witnessing the attack and/or collapse of the WTC; financial, safety, immigration concerns	Eating more or less than one had prior to September 11th and beginning or increasing use of prescription drugs
Thiel de Bocanegra et al., 2005	Longitudinal	65	8 months and 18 months	PCL-C	No PTSD clusters: 8 mth 5% 18mth 16%. At 18-month assessment: mean score 18.4 (SD 12.5) (max 68) PTSD diagnosis (all three clusters met) 8-month 21%; 18-month 27%	Those who met diagnostic criteria of PTSD at both time points were significantly younger at immigration	
Thiel de Bocanegra et al., 2006	Cross-sectional	139	18 months	PCL-C	42% PTSD by total score but out of 17 or higher (on scale of 1 to 66); 19% by meeting or exceeding cutoff for all three clusters		Higher mean PCL for those who (1) received prescription drugs after 9/11 vs those who did not (2) increased medication usage vs. those who did not (3) more likely to report that people said they changed after 9/11.
Wang et al., 2019	Longitudinal	26,723 1712 Self-identified Asian Americans	W1–W3	PTSD Checklist PCL-17	PTSD prevalence among Asians was 15.1 and 14.4% among Whites.		Job loss since 9/11 was significantly associated with an increased in the odds of long term PTSD (AOR = 1.8; 95% CI 1.9-2.7).
Welch et al., 2016	Longitudinal	17,062	Three time points: 2003-04; 2006-07; 2011-12;	PCL-17 Civilian		For worsening trajectory groups: not college graduate, unemployment, low social integration, unmet mental healthcare needs, 9/11-related job loss, witnessing 3+ horrific events, threat of 9/11 injury/death, 9/11-related bereavement, 9/11 injury, poor quality of life	
Wyka et al., 2019	Longitudinal	*N* = 25,143 *n* = 12,745 survivors	W1–W3	PTSD Checklist PCL-17	This paper examined the course of Probable PTSD and LRS among community members and responders to identify the stability and co-occurrence between them. Chronic PTSD (all 3 waves) was demonstrated in 55% of the community members		9/11 exposures such as dust cloud exposure and home damage mediated PTSD at w1 and w2, and marginally at w3 among community members.
Yu et al., 2019	Longitudinal	6377 Enrollees	Health and Employment Survey In Depth Study, 2017–2018	PTSD Checklist PCL-17			PTSD and 3 or more chronic health conditions increased odds of early retirement in a dose response manner, compared, not seen for those enrollees who did not have PTSD. OR for 3 or more chronic diseases and PTSD was 3.4 (2.4–4.7).

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
