# Peer review of "Post-Traumatic Stress Disorder among Survivors of the September 11, 2001 World Trade Center Attacks: A Review of the Literature"

_ijerph, 2020, doi:10.3390/ijerph17124344_

Round 1

Reviewer 1 Report

This is a careful review of PTSD among WTC survivors. 

I would suggest some very minor clarifications/edits: 

Line 104 - some first responders and volunteers were directly exposed to the attacks and onsite as the towers fell. This distincition is not clear as to a greater impact of psychological exposure for survivors versus some responders.

There should be more clarification regarding exposure to the WTC attack - what are they describing as this exposure? is it just seeing the planes hitting or is it seeing the towers fall? 

There is a typographical error in the table - Deckel et al it should say higher for women not high

Overall, this is a well written, well research review that adds to the growing body of literature regarding short and long term PTSD from a terrorist attack. The authors suggestions for future research needs are excellent.

Author Response

Reviewer 1:

This is a careful review of PTSD among WTC survivors. I would suggest some very minor clarifications/edits: Line 104 - some first responders and volunteers were directly exposed to the attacks and onsite as the towers fell. This distincition is not clear as to a greater impact of psychological exposure for survivors versus some responders.

We would like to thank the reviewer the pointing out this discrepancy. We have added additional details to lines 104-105 to make this point clearer.

There should be more clarification regarding exposure to the WTC attack - what are they describing as this exposure? is it just seeing the planes hitting or is it seeing the towers fall? 

We have addressed this concern in lines 106-107 by adding more details about the exposures and what those looked like for survivors. Additional details are also provided based on the exposure criteria used in each study (when provided) throughout the findings section.

There is a typographical error in the table - Deckel et al it should say higher for women not high

We have addressed this error and corrected the typographical issue in the table. We have also addressed additional spacing issues that were identified thanks to this comment.

Reviewer 2 Report

The paper, which consists of a systematic review,  adds to our knowledge a good summary of existing follow up research on the adult survivors of 9/11 who were not rescue workers in terms of their PTSD sysmptoms and related somatic conditions. 

Some of the findings are surprising, such that women experience a higher level of PTSD than men, and require further explanation which is not provided.

Rightly the authors indicate that a mixed quantitative and qualitative methods is required for this type of research, which the studies they have reviewed have not provided. Although they mention that issues relating to the social background require attention, it would seem that these studies have not provided sufficient in-depth research to this component.

The authors focus on the diagnosis and the symptoms that come with it. While they indicate the need to pay attention to mediating mechanisms, they do not say much about this element, or about existing interventions which have been applied to the population diagnosed with PTSD.

The paper would have been enriched by reviewing research on interventions, including the Post Traumatic Growth approach, which developed in particular at the post 9/11 period.  

Author Response

Reviewer 2:

The paper, which consists of a systematic review, adds to our knowledge a good summary of existing follow up research on the adult survivors of 9/11 who were not rescue workers in terms of their PTSD sysmptoms and related somatic conditions. 

We would like to thank the reviewer for their feedback and commentary on our manuscript.

Some of the findings are surprising, such that women experience a higher level of PTSD than men and require further explanation which is not provided.

We agree that the findings surrounding women versus men and their rates of PTSD post-9/11 were surprising, particularly given the varying degrees to which they were reported in these studies. To address these discrepancies and present some alternative explanations for why this may be the case we have included a paragraph in the discussion (lines 435-455).

Rightly the authors indicate that a mixed quantitative and qualitative methods is required for this type of research, which the studies they have reviewed have not provided. Although they mention that issues relating to the social background require attention, it would seem that these studies have not provided sufficient in-depth research to this component.

We agree with the reviewer that the majority of these studies do not provide sufficient detail or specified research focus on the areas of social background or barriers. As such, we present this point as a discussion item and suggest that future studies focus more attention on these factors as potentially salient influences in the experience of PTSD for survivors post-9/11.

The authors focus on the diagnosis and the symptoms that come with it. While they indicate the need to pay attention to mediating mechanisms, they do not say much about this element, or about existing interventions which have been applied to the population diagnosed with PTSD. The paper would have been enriched by reviewing research on interventions, including the Post Traumatic Growth approach, which developed in particular at the post 9/11 period.  

We appreciate the reviewer’s comments on this aspect of the literature pertaining to PTSD. For the purposes of this review our focus was to determine the current research that surrounded PTSD symptoms and co-morbidities among 9/11 survivors. We acknowledge the importance of interventions the implications they may have for improving PTSD symptoms among survivors. For this manuscript, though, we felt that the inclusion of interventions would have extended beyond the reach of this paper. We would be interested, however, in addressing PTSD interventions among survivors in a separate systematic review.

Reviewer 3 Report

  • Altamore, F., Grappasonni, I., Laxhman, N., Scuri, S., Petrelli, F., Grifantini, G., Accaramboni, P., Priebe, S.

    Psychological symptoms and quality of life after repeated exposure to earthquake: A cohort study in Italy
    (2020) PLoS ONE, 15 (5), art. no. e0233172, .

    Line 128; Regarding Methods, an evidence-based minimum set of items for reporting in reviews, as for example PRISMA of systematic reviews, has been applied?
  • Line 143; Figure 1: it is suggested to improve the image quality because the content shown in the boxes is not always legible
  • I suggest adding in the references a new interesting paper 

    Altamore, F., Grappasonni, I., Laxhman, N., Scuri, S., Petrelli, F., Grifantini, G., Accaramboni, P., Priebe, S.

    Psychological symptoms and quality of life after repeated exposure to earthquake: A cohort study in Italy
    (2020) PLoS ONE, 15 (5), art. no. e0233172, .

Author Response

Reviewer 3:

Line 128; Regarding Methods, an evidence-based minimum set of items for reporting in reviews, as for example PRISMA of systematic reviews, has been applied?

To the best of our abilities we attempted to use a PRISMA model for conducting the review of the literature. We have added additional details in figure one to be more inclusive of the PRISMA model for sample selection.

Line 143; Figure 1: it is suggested to improve the image quality because the content shown in the boxes is not always legible

We appreciate the reviewer’s feedback on the image quality. We have attempted to adjust the image, making it larger and more legible, as well as added the additional markers on the side to indicate the selection criteria phases as referenced in the above comment.

I suggest adding in the references a new interesting paper  Altamore, F., Grappasonni, I., Laxhman, N., Scuri, S., Petrelli, F., Grifantini, G., Accaramboni, P., Priebe, S. Psychological symptoms and quality of life after repeated exposure to earthquake: A cohort study in Italy (2020) PLoS ONE, 15 (5), art. no. e0233172.

We appreciate the reviewer’s suggestion to add this additional reference to the manuscript. We have added this reference in the introduction of the document in the paragraph beginning at line 71 through line 87 on PTSD Prevalence and Risk Factors.